# Individual Variabilities in Adipose Stem Cell Proliferation, Gene Expression and Responses to Lipopolysaccharide Stimulation

**DOI:** 10.3390/ijms232012534

**Published:** 2022-10-19

**Authors:** Rumana Yasmeen, Quynhchi Pham, Naomi K. Fukagawa, Thomas T. Y. Wang

**Affiliations:** 1Diet, Genomics and Immunology Laboratory, Beltsville Human Nutrition Research Center, Agricultural Research Service, United States Department of Agriculture, Beltsville, MD 20705, USA; 2Division of Food Labeling & Standards, Office of Nutrition and Food Labeling, Center for Food Safety and Applied Nutrition, U.S. Food and Drug Administration, College Park, MD 20740, USA

**Keywords:** adipose stem cell, inflammation, individual differences, sex, LPS

## Abstract

Adipose stem cells (ASCs) are reported to play a role in normal physiology as well as in inflammation and disease. The objective of this work was to elucidate inter-individual differences in growth, gene expression and response to inflammatory stimuli in ASCs from different donors. Human ASC1 (male donor) and ASC2 (female donor) were purchased from Lonza (Walkersville, MD). Cell proliferation was determined by the sulforhodamine B assay. After time-dependent treatment of ASCs with or without bacterial lipopolysaccharide (LPS), marker gene mRNAs for proliferation, steroid hormones, and xenobiotic and immune pathways were determined using RT-PCR, and secreted cytokine levels in media were measured using the Bio-Plex cytokine assay kit. ASCs from both donors expressed androgen receptors but not estrogen receptors. ASC2 had a 2-fold higher proliferation rate and a 6-fold higher level of proliferation marker Ki67 mRNA than ASC1. ASC2 exhibited significantly greater fold induction of TNF-α and CCL2 by LPS compared to ASC1. TNF-α and GM-CSF protein levels were also significantly higher in the LPS-induced ASC2 media, but IL-6 secretion was higher in the LPS-induced ASC1 media. Our findings suggest that inter-individual variability and/or possible sex differences exist in ASCs, which may serve as a key determinant to inflammatory responses of ASCs.

## 1. Introduction

Recent advances in the biological sciences have led to the emergent concepts of precision medicine and/or personalized medicine [1,2,3,4]. However, understanding the distinct differences between individuals, as well as genetic variation in susceptibility to diseases such as obesity, diabetes and cancers, remains a daunting task. Comprehensive cataloguing and documentation of individual differences at, for example, the gene, cell and various omics levels are necessary to fully realize the goal of personalized health care.

Stem cells are undifferentiated cells that are capable of self-renewal and differentiation into multiple cell lineages [5,6]. In in vitro tissue engineering, cell differentiation can be induced by growing cells on scaffolds with appropriate composition, architecture and physicochemical and mechanical properties [7,8]. Among adult stem cells, adipose stem cells (ASCs) seem to be the most advantageous for use in cell therapies, tissue engineering and regenerative medicine. ASCs are abundant in adipose tissue and can be easily isolated in large quantities [9,10]. Moreover, due to adipose tissue’s subcutaneous localization, it is relatively easy to access ASCs without significant donor site morbidity [11,12]. It has been reported that 300 times more stem cells can be obtained from adipose tissue than from equal amounts of bone marrow [12]. Hence, ASCs have become attractive candidates for multiple purposes. Yet, how individual differences may influence the utility of ASCs remains unclear.

Although current attention and interest often revolve around utilizing ASCs for tissue engineering, these cells actually play critical roles in normal physiology as well as in the pathogenesis of specific conditions in an individual [13]. For example, ASCs regulate adipose tissue homeostasis through their ability to differentiate into different lineages due to their plastic nature [14]. These processes are accomplished by expression of different growth factors and receptors and secretion of cytokines and different mediators [15,16]. In addition, ASCs are responsible, at least in part, for adipose tissue depot-related heterogeneity [17,18]. ASC function and plasticity are also closely associated with inflammation [19,20]. In addition to the ability to secrete inflammatory cytokines such as IL-1, TNF-α and IL-6 [21,22], ASCs also respond to inflammatory stimuli such as bacterial LPS and TNF-α [22,23]. In pathophysiological states such as obesity the proliferation and differentiation of ASCs are influenced by pro-inflammatory cytokines [24,25]. It is speculated that ASCs can contribute to obesity-associated inflammation and metabolic disorders [26,27]. Furthermore, individual differences such as age and sex may also influence ASC responses and functions [28,29]. In addition to obesity and aging, ASCs are reported to be involved in pathogenesis of other diseases. For example, the development, progression and metastasis of cancer in breast, prostate and lung are thought to be influenced by ASCs through their secretion of cytokines and chemokines [30,31,32,33,34,35]. Others have documented the attraction of ASCs by tumor cells, as well as attraction of M2-like tumor-associated macrophages by ASCs, as mechanisms by which ASCs promote carcinogenesis [36,37]. Keeping in mind the tumor-homing ability of ASCs, some have utilized ASCs as a “trojan horse” to deliver chemotherapeutic drugs into tumor cells [38]. However, the various biological effects exerted by an individual’s ASCs remain unclear, and they warrant further elucidation, especially in the age of precision medicine.

To further contribute to the effort in personalized medicine and to address the deficiency in the literature, the goals of the current study are (1) to document potential individual differences in adipose stem cells and (2) to elucidate potential pathways related to growth, differentiation, response to inflammatory stimuli and interaction with other stromal cells. The current study compared adipose stem cells from two different donors and described the growth, differentiation properties as well as differential responses to inflammatory stimuli and effects on immune cells. Our results support the concept that ASCs are distinct and that the individualism of ASCs need to be taken into consideration when implementing their application in the treatment of specific conditions, assessing individual’s responses to inflammation and relationship with diseases risks. 

## 2. Results

### 2.1. Preliminary Characterization of Adipose Stem Cells

The sources of the two ASCs used in this study were from two different individuals. ASC1 was derived from a healthy black male subject, age 33, BMI 25. ASC2 was from a healthy black female subject, age 37, BMI 29. Due to the gender differences, we assessed mRNA expression levels of the sex steroid hormone receptors AR, ESR1 and ESR2 to gain additional baseline information on these cells. Interestingly, we found that under the same assaying conditions ESR1 and ESR2 were below detection limits in both ASCs. However, both ASCs were found to express similar levels of AR mRNA (Appendix A). 

### 2.2. Comparison of Cell Proliferative Capacity between ASCs

Cell growth is a global endpoint for cell health; therefore, we first assessed the growth rates of the individual ASCs. We found individual ASCs significantly differed in their growth rate (Figure 1a). ASC2 appeared to have a significantly higher growth rate than ASC1. Consistent with this, the faster growing ASC2 also expressed significantly higher levels (~7 fold) of the proliferation marker Ki67 than the slower growing ASC1 (Figure 1b). In addition, the plating efficiency appeared to be different. With the same number of cells (1.5 × 10^5^/mL) plated for the experiments, there appeared to be ~2 fold (*p* = 0.0058) more ASC2 cells attached to the plate than ASC1 cells 24 h after plating before the start of the proliferation experiments (Figure 1a).

### 2.3. Comparison of Adipogenic Differentiation Potential between ASCs 

Since ASCs are multipotent cells that are capable of differentiating into multiple lineages, we next examined the differentiation potential of the ASCs into adipocytes using lipid accumulation as a terminal indicator. As shown in Figure 2a, upon differentiation initiation ASC1 appeared to accumulate significantly higher levels of lipid than ASC2. Furthermore, we compared selected differentiated adipocyte markers PPAR-gamma and FABP4 mRNA expression in the ASCs. PPAR-gamma and FABP4 were significantly up-regulated upon initiation of differentiation of the ASCs into adipocytes (Figure 2b,c). More importantly, the magnitude of up-regulation of mRNA of these genes was significantly higher in ASC1 than ASC2.

### 2.4. Comparison of LPS-Induced Changes in Cytokine mRNA and Protein Expression in ASCs 

ASCs have been reported to be more sensitive than adipocytes in secreting inflammatory cytokines in response to inflammatory stimuli such as bacterial LPS [39]. Hence, we compared the temporal effects of cytokine and chemokine mRNA levels changes when ASC1 and ASC2 were treated with LPS (Figure 3). Exposure to LPS induced mRNA expression of several cytokines and chemokines. Interestingly, the temporal effects of the induction of cytokines/chemokines mRNA appeared to differ among those measured. In all cases the rapid increases in mRNA levels with LPS exposure that peaked at 2 or 4 h depended on specific cytokines/chemokines. TNF-α and GM-CSF in particular (Figure 3e,f) had very sharp declines after peaking. For IL-1β, IL-6 and IL-8 (Figure 3a–c) the decline after peak appeared to occur more slowly during the period examined. The pattern of cytokines/chemokines responses to LPS exposure appeared to differ between ASC1 (male donor) and ASC2 (female donor) (Figure 3). The cytokines/chemokines mRNA responses of ASC2 toward LPS appeared to be consistently higher than those of ASC1. Notably, CCL2, a monocyte attracting chemokine, was induced ~4-fold higher in ASC2 than ASC1 after 8 h exposure to LPS. Additionally, the mRNA levels of CCL2 (Figure 3d) remained high after stimulation, especially for ASC2.

Secretion of selected pro-inflammatory cytokines/chemokines at the protein level was also determined. Both ASCs responded to LPS stimulation and secreted pro-inflammatory cytokines/chemokines. TNF-α, IL-6, IL-8 and GM-CSF were secreted at higher levels (Figure 4). In particular, ASC1 appeared to secrete higher levels of IL-6 (Figure 4a) at baseline (*p* < 0.0001) as well as in response to LPS stimulation (*p* < 0.0001) than ASC2. ASC1 secreted higher levels of IL-8 (Figure 4b) at earlier time points than ASC2 (*p* < 0.0001). ASC2 appeared to secrete significantly higher levels of TNF-α and GM-CSF than ASC1 (Figure 4c,d, *p* < 0.0001). Additional cytokines/chemokines, including IL-2, 4, 10 and IFN-γ, were examined. However, IL-10 was below detection limit under the same assay conditions. IL-2, IL-4 and IFN-γ concentrations (Appendix A) were much lower than those illustrated in Figure 4. These cytokines/chemokines were found to be also inducible by LPS. IL-2 was detected at <10 pg/mL, and we did not observe differences in IL-2, in terms of temporal responses to LPS, between ASC1 and ASC2. However, the baseline level of IL-2 appeared to be higher in ASC1 (*p* < 0.001). IL-4 was detected at <10 pg/mL, and the response patterns appeared to be similar to those of GM-CSF. IFN-γ detection was <20 pg/mL and response patterns appeared to be similar to those of IL-2. 

### 2.5. Effects of ASC-Derived Conditioned Media on Human THP-1 Cell Differentiation 

As shown above, ASCs from male and female donors appeared to differentially produce several cytokines/chemokines. The differences in GM-CSF protein, a known regulator of monocyte differentiation to macrophage [40], was found to differ the most in terms of magnitude. We sought to confirm whether such differences in GM-CSF production may be sufficient to elicit differences in the differentiation of monocytes. As illustrated in Figure 5, conditioned media from both ASC1 and ASC2 induced an increase in the mRNA levels of the macrophage differentiation marker gene, CD14, in the THP-1 human monocytic cell line after 48 h incubation. Consistent with cytokine data reported above, ASC2-derived conditioned media appeared to elicit a greater increase in CD14 mRNA levels (~12 fold) than conditioned media derived from ASC1 (~8 fold). However, we did not observe adherence of THP-1 to the culture plate under the conditioned media treatment, as stimulation with PMA is necessary to elicit the adherent phenotype of THP-1. 

## 3. Discussion

The present study provided biological and molecular documentation to confirm our hypothesis that individuals’ ASCs may be very different. We observed significant differences between two ASCs in (1) growth rate, (2) ability to differentiate into adipocytes, (3) responses toward inflammatory stimuli, and (4) interaction with macrophages. Our data also highlight the potentially important effect of sex on the responses analyzed. 

Comparison of our ASC1 and ASC2 data supports the notion that individual differences exist at the ASC level. Interestingly, ASC1, which grew more slowly, appeared to be more readily differentiated into adipocytes, as supported by lower Ki67 levels, increased lipid accumulation and induction of adipocyte markers PPAR-gamma and FABP4. In contrast, ASC2 was just the opposite with ~7-fold higher Ki67 mRNA levels, little lipid accumulation and much lower PPAR-gamma and FABP4 expression after initiating differentiation. These data suggest that the regulation of various proliferation/differentiation pathways differs in these cells as manifested by the differences in the phenotypic display of ASC1 and ASC2. Though we do not know what proximal pathways may be different between ASC1 and ASC2, the data are reflective of individual differences in the adipose stem cells when similarly grown and stimulated. The differences we observed at the individual level may significantly impact the physiology as well as susceptibility to different diseases. For example, development of obesity is related to adipocyte size (hypertrophy) and number (hyperplasia) [41]. One may reason those individuals with the ASC1-like phenotype, because of its lower proliferation rate, may have a lower number of adipocytes and a lesser propensity to become obese. In contrast, those with the ASC2-like phenotype may be more susceptible to obesity as there will be potentially more adipocytes generated from this phenotype. Alternatively, individuals with the ASC1 phenotype, which takes up lipid more rapidly, may be more susceptible. Further studies are necessary to elucidate/validate the physiological impact of such differences. 

The two ASCs studied also showed differential responses toward LPS stimulation. In general, ASC2 appeared to be more responsive toward LPS, based on the magnitude of changes elicited by LPS stimulation on the production of chemokines and cytokines. In particular, secretion of GM-CSF, a macrophage differentiation factor, appeared to be significantly different between the two ASCs. We also demonstrated that these differences were reflected at biological levels. Culturing human monocytic THP-1 cells in conditioned media from ASC2 led to higher induction of monocyte/macrophage differentiation marker CD14 mRNA levels, supporting the notion that higher production of GM-CSF by ASC2 may contribute to ASC2 being a more potent elicitor of monocyte/macrophage differentiation. This finding also supports the notion that exposure to bacteria or other inflammatory stimuli may thus elicit differential immune responses in individuals, depending on their ASC phenotype. We reason those individuals with the ASC2-like phenotype, which generally exhibited higher cytokine output, may be more susceptible to chronic low-level inflammation and development of obesity-related risk factors and cancers. Interestingly, we did not observe THP-1 cells become adherent to the culture plate, suggesting that additional signals may be necessary to display such phenotype.

One discrepancy in our data is related to IL-6, where the protein and mRNA data are not consistent. Induction of IL-6 mRNA was higher in ASC2 than ASC1, but the opposite was observed for protein, with higher levels of IL-6 protein detected in the media harvested from ASC1 than that from ASC2. This may indicate differential translational and transcriptional regulation in IL-6, but it is unclear how this occurred and warrants further study. Significantly, higher IL-6 protein was detected in ASC1-conditioned media compared to ASC2-conditioned media. Differential production of IL-6 by ASCs in the unstimulated baseline may be of interest. It has been reported that the presence of IL-6 appears to play a role in driving monocytic cell differentiation toward macrophages rather than dendritic cells [42]. Therefore, it is possible that different individuals, based on the ability of their ASCs to produce IL-6, may elicit differential inflammatory responses and lead to differences in disease consequences. For example, inhibition of dendritic cells appears to be a mechanism by which tumor cells escape immune surveillance [43]. It is possible that a person with ASCs that produce high amounts of IL-6 may be at a greater risk for cancer due to a shift in immune cells toward macrophages rather than dendritic cells. However, due to complexities in IL-6-related immune regulation, as well as the interaction with other cytokines/chemokines such as GM-CSF or M-CSF [42,43], this hypothesis requires considerable experimentation and validation.

One intriguing difference we would like to point out is that ASC1 was from a male donor and ASC2 was from a female donor. Sex differences in diseases related to the immune system exist [44]. Given that there are such differences in growth, differentiation and responses to LPS, we speculate that perhaps the ASC phenotype may be sex dependent. If so, sex-dependent susceptibility to obesity, infection or other diseases may be related to an individual’s ASCs. For example, females (ASC2) may be more prone to becoming obese because there may be more adipocytes in the adipose tissue of women. Additionally, females may have higher risk for adipose-related sublevel inflammation because ASC2 appeared to produce more inflammation-related chemokines/cytokines upon stimulation by LPS. However, more detailed/extensive analyses are necessary to support our hypothesis. Another interesting but puzzling observation was the lack of expression of the estrogen receptors (ESR1 and ESR2) in both ASC1 and ASC2, although they were derived from different sexes. In contrast, ASCs expressed similar levels of AR. It is unclear whether in their natural adipose tissue environment, i.e., in vivo, the ASCs would express patterns similar to the in vitro results reported in our study. If so, the regulation of ASCs in the adipose tissue will need to be examined and warrants further study.

Adipose tissues are easily accessible and large numbers of adipose stem cells can be obtained from the harvested tissue [11,12]. We hypothesized that assessment of adipose stem cell growth, immune responses, as well as gene markers, may provide a potential tool to define an individual’s responsiveness to inflammatory stressors. However, more work is necessary to validate this hypothesis.

In summary, in support of personalized medicine/nutrition, we found that an individual’s ASCs may be very different in terms of growth, differentiation and responses toward inflammatory stimuli. We propose that the individual ASC phenotype may affect an individual’s responses to inflammation as well as susceptibility toward disease such as obesity, cancers. The findings of this study open up several exciting avenues of research, but further work is necessary to elucidate and validate the hypothesis that ASCs may play a critical role in modulation of an individual’s immune responses and disease risk.

## 4. Materials and Methods

### 4.1. Chemical and Reagents 

Human adipose stem cells (ASCs) from healthy donors (one male donor 29635, Lot no. 0000535975 and one female donor 27922, Lot no. 0000430174) were purchased from Lonza (Cat #PT-5006, Walkersville, MD, USA). DMEM/F-12 (Cat #11320-033), supplemented with 10% fetal bovine serum (FBS), 100 U/mL penicillin, and 100 mg/mL streptomycin, were all obtained from ThermoFisher Scientific (Grand Island, NY, USA). Lipopolysaccharide (LPS), phorbol-12-myristate-13-acetate (PMA) and gelatin were purchased from Sigma (St. Louis, MO, USA). Bioplex Pro-Human Cytokine 8-Plex assay (Cat #M50000007A) was purchased from Bio-Rad Laboratories (Hercules, CA, USA). PGM-2^TM^ Preadipocyte Growth Medium-2 BulletKit^TM^ (Cat #PT-8002) and Oil Red O (Cat #PT-7009) were obtained from Lonza (Walkersville, MD, USA). TRIzol reagent, TaqMan Fast Universal PCR Master Mix, and TaqMan real-time PCR primers and probes were obtained from Life Technologies/ThermoFisher Scientific (Waltham, MA, USA). All chemicals were analytical reagent grade. The following Gene Expression Assay primers were purchased from ThermoFisher Scientific (Grand Island, NY, USA): TBP (Hs00427620_m1), Ki67 (Hs01032443_m1), IL-6 (Hs00985639_m1), IL-1 (01555410_m1), IL-8 (Hs00174103_m1), TNF-α (HS00174128_m1), CCL2 (Hs00234140_m1), CD14 (Hs02621496_m1), PPAR-gamma (Hs01115513_m1), FABP4 (Hs01086177_m1), and CSF-2 (Hs99999044_m1).

### 4.2. Cell Culture 

ASCs were cultured in DMEM/F-12 medium containing 10% FBS, 1% penicillin-streptomycin. Cells were routinely grown to confluency in T-75 flasks (ThermoFisher Scientific, Grand Island, NY, USA) and used as stock for experiments conducted and culture medium-conditioned for specific experiments as described in their respective sections.

### 4.3. Cytokines/Chemokines Analysis

ASCs (1.5 × 10^5^/mL) were plated overnight in 6-well plates (ThermoFisher Scientific). Cells were then stimulated with LPS (10 ng/mL) and supernatants harvested at 4, 6, 8, and 24 h for Bio-Plex (Bio-Rad Laboratories, Hercules, CA, USA) cytokines/chemokines analysis. Briefly, cytokine and chemokine signaling molecules in supernatants of ASCs treated under various conditions were measured using magnetic bead-based assays with commercial kits (Bio-Rad Laboratories, Hercules, CA, USA). The assays were performed according to the manufacturer’s protocol on a Bio-Plex 200 system with Bio-Plex Pro II Wash station. Data were acquired and analyzed using Bio-Plex Manager Software v6.1. Adipocyte differentiation was carried out using PGM-2^TM^ Preadipocyte Growth Medium-2 BulletKit^TM^ (Lonza, Walkersville, MD, USA) according to the manufacturer’s detailed protocol. 

### 4.4. Cell Proliferation Assay

ASCs at a density of 1.5 × 10^5^/mL were plated overnight in 24-well plates and harvested at 0, 24, 48 and 72 h. Media was aspirated and replaced with fresh media every 24 h throughout the duration of the experiment. Total protein was measured for each time point using the sulforhodamine B (SRB) assay [45]. Briefly, at the desired time point, medium was aspirated and cells then fixed with 10% Trichloroacetic acid (TCA) for 1 h at 4 °C. TCA was removed and wells were washed 5 times with deionized water and air-dried at room temperature. Subsequently, 0.4% SRB in 1% acetic acid was added and incubated for 20 min before wells were washed with 1% acetic acid. Bound dye was dissolved with 10 mM unbuffered Tris-base (pH 10.5) for 5 min and read at optical density between 490–530 nm on SpectroMax Plus (Molecular Devices, San Jose, CA, USA).

### 4.5. Oil Red O Staining

ASCs at a density of 1.0 × 10^5^/mL were plated in 24-well plates pre-coated with 0.1% gelatin (Sigma, Cat #G1393) according to manufacturer’s instructions to prevent cells from detaching. Adipogenic differentiation was induced following specific procedural instructions from Lonza PGM-2^TM^ Preadipocyte Growth Medium-2 BulletKit^TM^. Differentiation was carried out until day 9 with medium changed every other day. Adipocytes were then fixed with 10% buffered formalin for 1 h, washed and stained with Oil Red O (Lonza, Cat# PT-7009) according to the manufacturer’s instructions. Briefly, adipocytes were fixed at room temperature for 20 min, then wells were washed 4 times with deionized water. Lipids were quantified by dissolving in 100% isopropanol and reading absorbance at 500 nm using Molecular Devices SpectroMax Plus. 

### 4.6. Real-Time PCR Analysis of Gene Expression

ASCs (1.5 × 10^5^/mL) were plated overnight in 6-well plates (ThermoFisher Scientific). Cells were then stimulated with LPS (10 ng/mL) and harvested at 0, 2, 4, 6, and 8 h for real-time PCR (RT-PCR) analysis. Total RNA was isolated using TRIzol reagent (ThermoFisher Scientific, Grand Island, NY, USA) and cDNA was synthesized using AffinityScript Multiple Temperature cDNA Synthesis kit (Agilent Technologies, Santa Clara, CA, USA) as previously described (Need Reference). RT-PCR was carried out using a TaqMan Fast Universal PCR Master Mix on a 7900HT FAST real-time PCR System (Applied Biosystems, Foster City, CA, USA). Relative mRNA expression values were generated using ΔΔ*C*T method (Applied Biosystems, Foster City, CA, USA). Human TATA–box binding protein (TBP-Hs00427620_m1) was used as an endogenous control for all gene expression normalization analysis.

### 4.7. Conditioned Media Treatment of Human THP-1 Cells 

ASCs at a density of 1.5 × 10^5^/mL were plated overnight in T75 flask and cells were then stimulated with LPS (10 ng/mL) and conditioned media (CM) harvested after 24 h in culture. Control media were obtained by following the same procedure as CM but without ASCs in the flask. To assess the differentiation potential of conditioned media, human monocytic THP-1 cells were plated in control or conditioned media and cells were harvested after 48 h incubation in the respective media. Cells were harvested after 48 h for total RNA isolation and the differentiation marker CD14 mRNA levels were determined using RT-PCR.

### 4.8. Statistics

GraphPad Prism 9 (GraphPad Software, La Jolla, CA, USA) was used for statistical analysis of the data obtained. Growth curves were compared using simple linear regression analysis. Time-course data for gene expression and cytokines/chemokines levels were analyzed using regression analysis and by two-way ANOVA followed by Tukey’s post hoc test to determine time, treatment and time x treatment effects. Time-course curves were also analyzed for area under the curve (AUC) followed by one-way ANOVA. Two-way ANOVA, followed by Tukey’s post hoc test, was used for multi-group analysis. Student’s *t*-test was used for two-group comparisons. Results are expressed as mean ± SD. A *p* value ≤ 0.05 was considered significant.

## 5. Conclusions

Individual differences in adipose stem cells’ growth, function and responses toward inflammation may influence disease outcomes and may serve as a tool to define individual susceptibility to diseases and infection.

## Figures and Tables

**Figure 1 ijms-23-12534-f001:**
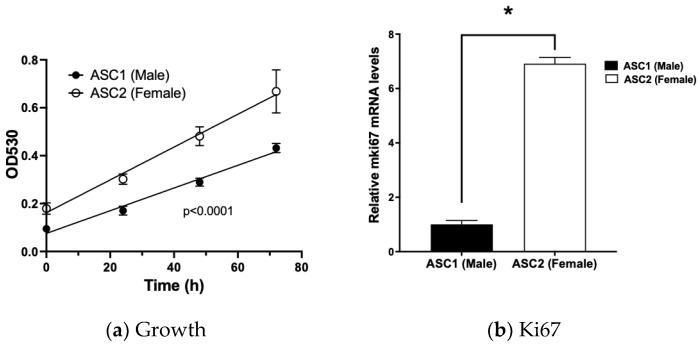
Differential growth characteristics of individual adipose stem cells (ASCs). (**a**) Comparison of ASC growth rates. ASCs were cultured and cell growth determined as described in Materials and Methods using the SRB method. Results are expressed as mean +/− SD (n = 8). Regression analysis was performed and the two curves were significantly different at *p* < 0.0001. (**b**) Comparison of proliferation marker Ki67 mRNA levels. ASCs were cultured and proliferation marker Ki67 mRNA levels determined as described in Materials and Methods. Data expressed as relative mRNA levels to ASC1 (Mean +/− SD, n = 3). * indicates significantly different from each other at *p* < 0.05.

**Figure 2 ijms-23-12534-f002:**
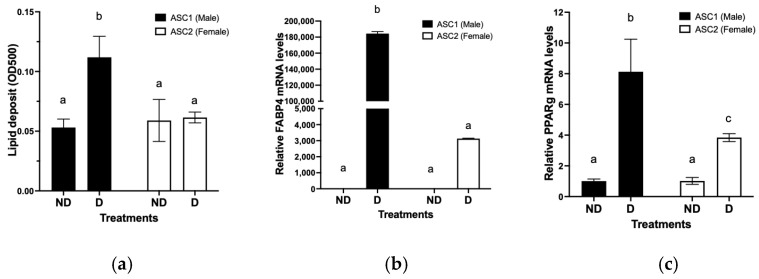
Differences in differentiation potential of individual adipose stem cells (ASCs). ASCs were cultured and differentiation potential was assessed by measuring Oil Red O staining of lipid deposit and differentiation marker genes PPAR-gamma and FABP4 mRNA levels as described in Materials and Methods. (**a**) Oil Red O staining of lipid deposit. Results are expressed as OD500 +/− SD (n = 3). (**b**) FABP4 mRNA levels. Results are expressed as relative mRNA levels (Mean +/− SD, n = 3) to non-differentiated ASC1. (**c**) PPAR-gamma mRNA levels. Results are expressed as relative mRNA levels (Mean +/− SD, n = 3) to non-differentiated ASC1. ND: Non-differentiated. D: Differentiated. Bars with different letter indicate significantly different at *p* < 0.05.

**Figure 3 ijms-23-12534-f003:**
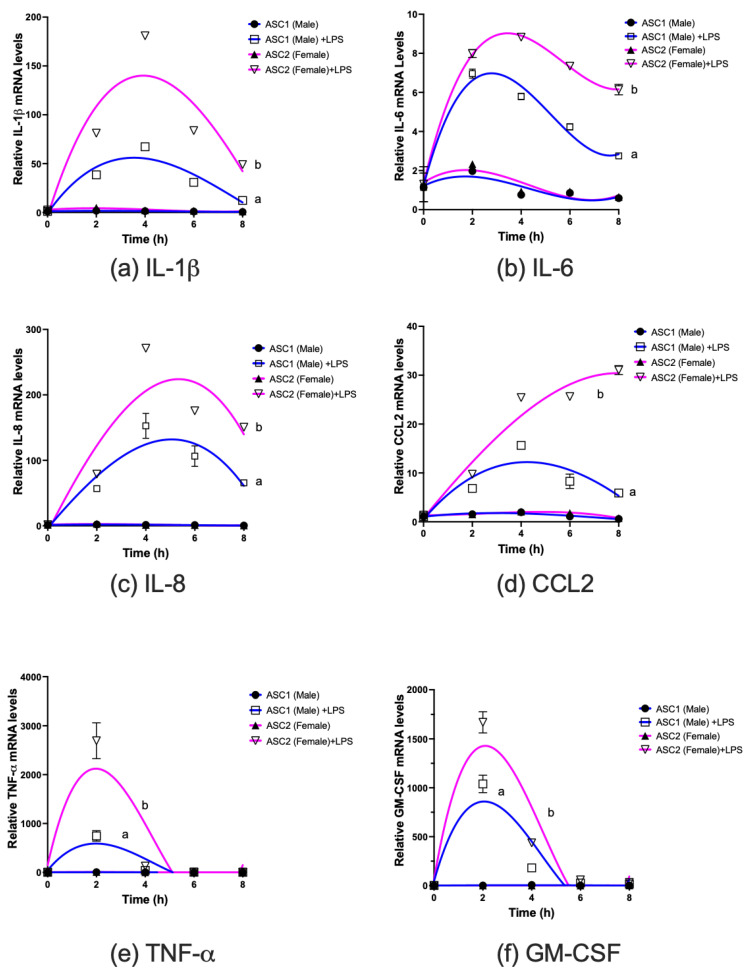
Individual adipose stem cells (ASCs) differentially induce cytokines mRNA expression in response to inflammatory stimuli. ASCs were cultured and treated with or without LPS (10 ng/mL), cells were harvested at different time points and total RNA was isolated as described in Materials and Methods. Selected inflammation-associated cytokines mRNA levels were determined using RT-PCR as described in Materials and Methods. Results expressed as relative mRNA levels (mean +/− SD, n = 3) to 0 time. Curves with different letter indicate significantly different at *p* < 0.05 or less.

**Figure 4 ijms-23-12534-f004:**
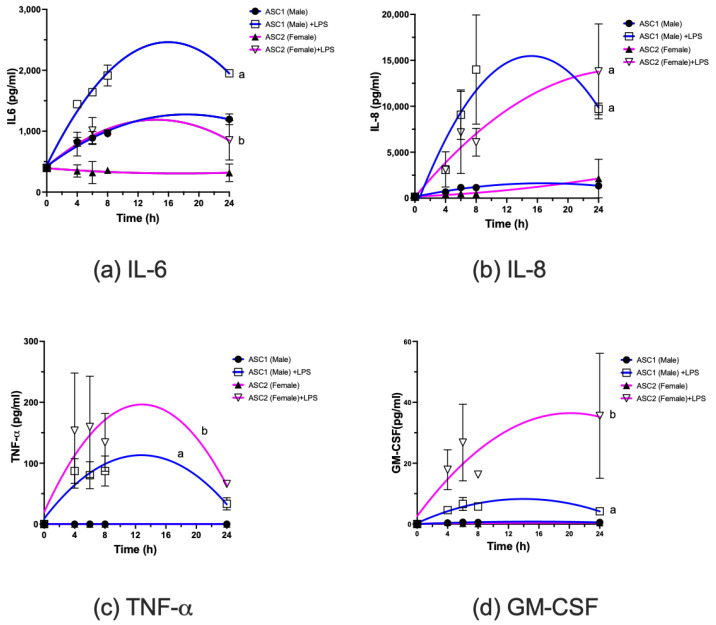
Individual adipose stem cells (ASCs) differentially secrete selected cytokines in response to inflammatory stimuli. ASCs were cultured and treated with or without LPS (10 ng/mL), media were harvested at different time points and selected cytokines protein levels determined as described in Materials and Methods. Results expressed as pg/mL (mean +/− SD, n = 4). Curves with different letter indicate significantly different at *p* < 0.05 or less.

**Figure 5 ijms-23-12534-f005:**
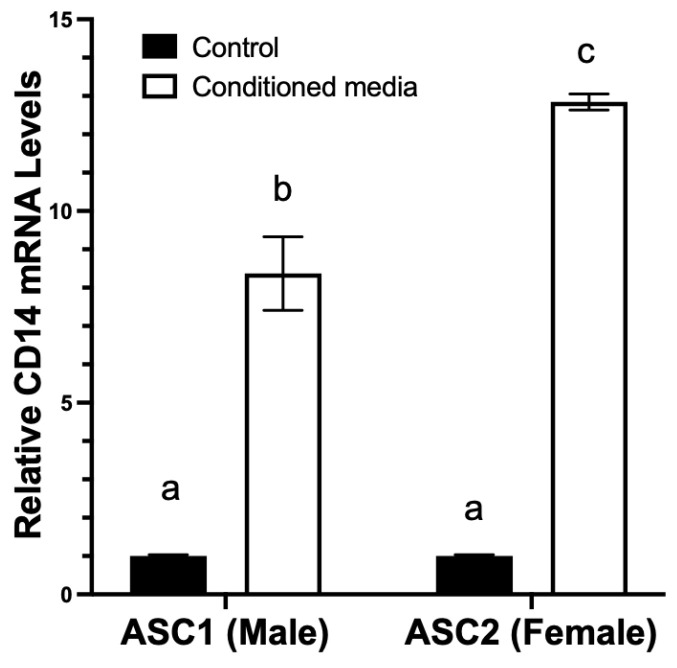
Effects of ASC-derived conditioned media on THP-1 cell differentiation. ASC-conditioned media were harvested as described in Materials and Methods. Human monocytic THP-1 cells were cultured in ASC-derived conditioned media for 48 h as described in Material and Methods. Cells were then harvested and total RNA isolated for determination of differentiation maker CD14 mRNA levels as described in Materials and Methods. Results are expressed as relative mRNA levels (Mean +/− SD, n = 3) to control media. Black bar: control media. White bar: conditioned media. Bars with different letters indicate significant differences at *p* < 0.05.

## Data Availability

The data presented in this study are available in article or Appendix A.

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
