# Peer review of "Individual Variabilities in Adipose Stem Cell Proliferation, Gene Expression and Responses to Lipopolysaccharide Stimulation"

_ijms, 2022, doi:10.3390/ijms232012534_

Round 1
Reviewer 1 Report
1. The references must be according to the journal's style.
2. Overall, the manuscript is interesting. It presents clear explications of the studies conducted, and the results are well explained. But, the authors of the manuscript must investigate even deeper the problem, and gather enough samples, conduct the tests, and finally be able to draw a conclusion. Only two pacients investigated, is not enough for a medical study.
3. Also, in the Introduction, it must be presented in a paragraph what is the purpose of the study and what the novelty of this work is.
Reviewer 2 Report
The authors investigated the inter-individual differences in growth, gene expression and response to inflammatory stimuli in ASC from male and female donors).They found the presence of individual differences in adipose stem cells’ growth, function and responses toward inflammation. This is particularly important because it could influence disease outcome and may serve as tool to define individual susceptibility to diseases and infection.
The manuscript is well written but there are some notes all over the text that should be corrected, in example line 84-85...etc
The main limitation of the study is that the authors investigated only one male cell line and one female cell line. to keep more robust the data they should include more cells, since they are stating that it is present an inter individual variability bewtween different subjects.
I would suggest to add more data before pubblication
Reviewer 3 Report
In the present manuscript by Yasmeen et al., demonstrates the Individual variabilities in adipose stem cell proliferation, differentiation and response to inflammatory signals. Authors use ASC from 2 different donors and analyze cell proliferation by sulforhodamine B assay and Ki67 expression. They analyzed their differentiation to adipocytes by analyzing lipid droplet accumulation, FABP4 and PPARg expression. Moreover, authors observed differences in responses to LPS stimulation. However interesting concept. But, the manuscript lacks additional supporting data to confirm their hypothesis (mostly data was conformed using mRNA expression). Critical details of experimental procedures and analysis were lacking. Moreover, the manuscript presented is poorly written, riddled with grammatical errors, and contains many confusing sentences that make it difficult to interpret the results and conclusions of the author. Some specific comments are outlined below.
1) Authors build up their entire conclusion regarding “inter-individual variability and sex differences exist in ASC” based on using 1 male and 1 female donor. But it does not confirm that the differences observed were due to sex or individual or the retrieval from lot no. of Cryopreserved ampule of Human Adipose-Derived Stem Cells from Lonza. To confirm their statement authors are required at least 3 males vs 3 females OR 3 individual donors.
2) In line 84, authors mention “…both ASC were found to express similar levels of AR mRNA but not the estrogen receptors” Both in abstract and result section but the related data is missing in the manuscript.
3) In all the figures what does a, b, c statistical difference signifies (what was the p value, which groups were compared). All the details should be mentioned in the legends.
4) In figure 2, legends section indicates PPAR-gamma FABP4 mRNA levels results are expressed as relative mRNA levels to ASC1. But it is not clear whether it is relative to differentiated or undifferentiated ASC1. Also, why specifically in this figure data were represented relative to ASC1?
5) It is not clear whether in figure 3 actual mRNA expression levels relative to internal control are shown OR the data is represented as relative fold change?
6) IL-6 mRNA level shown in figure 3b does not co-relate with protein level of IL-6 as shown in figure 4a.
7) How was the viability of ASC1 and ASC2 effected after acute and chronic treatment of LPS?
8) Though-out the result text for figure 4 authors mention about statistical difference observed in results but it is not reflected in the data points, meaning at which time point authors observed statistical difference?
9) In line 170-171 authors state that “ASC2 appeared to secrete significantly higher levels of TNF- and GM-CSF than ASC1 (Figure 4c, d, p<0.0001))” but the data does not support their statement.
10) There are number of lines that have unnecessary statements that need to be deleted like in line numbers 84-85, 119-120, 180, 206-207
11) Please remove hyphen from CD14 in the text.
12) Please remove “period” from the end of title of the manuscript.
Round 2
Reviewer 1 Report
The authors have answered the problems that were asked about their study. It would be very interesting if the authors continue their research and provide more-depth results and a follow-up study which would contribute to the scientific community.
Author Response
We thank the reviewer for constructive comment and hope to produce exciting follow up results in the near future.
Reviewer 3 Report
In the revised manuscript authors tried to address most of my comment. Authors need to perform additional experiments to support and justify their claim.
I still feel that the title of the manuscript is not justified. Using 1 male and 1 female to account for individual variabilities is the claim beyond providing the sufficient evidence. The differences observed may not be reflecting the individual variation but just the matter of sex.
Authors claim that “message level change is in good agreement with protein expression levels”. But again, it is a claim beyond provided evidence. Protein like PPARg have been well documented to be controlled at so many levels by posttranslational modification like phosphorylation at its serine residues. Even the cytokine mRNA expression of IL6 in this manuscript did not co-relate well with protein level.
Round 3
Reviewer 3 Report
In the present manuscript entitled "Individual variabilities in adipose stem cell proliferation, gene expression and responses to lipopolysaccharide stimulation" authors used ASC from 2 donor samples to propose/hypothesize that gender and ethnicity may be the variable factors that may ultimately lead to differential physiological consequences. I understand that in order to address sex, ethnic variables, will require extensive resources and personnel. But to draw the conclusion with 1 male and 1 female is a major leap for the claim. Authors need to thoroughly acknowledge this limitation of their study and include it in discussion. I believe including 1 more donor would be have at least strengthened their claim.